# Importance of Monitoring Fetal and Neonatal Vitality in Bovine Practices

**DOI:** 10.3390/ani13061081

**Published:** 2023-03-17

**Authors:** Ottó Szenci

**Affiliations:** Department of Obstetrics and Food Animal Medicine Clinic, University of Veterinary Medicine Budapest, Dóra Major, H-2225 Üllő, Hungary; szenci.otto@univet.hu

**Keywords:** dairy cow, fetal and neonatal vitality, acid–base and lactate measurement, ultrasonography, pulse oximetry

## Abstract

**Simple Summary:**

The assessment of fetal and neonatal vitality plays an essential role in preventing stillbirth in bovine cattle. Therefore, an accurate diagnosis of fetal vitality prior to obstetric assistance or neonatal vitality after birth is crucial for ensuring prompt and accurate treatment. If severe asphyxia occurs before the availability of obstetric assistance, a cesarean section can be performed as a less life-threatening alternative for the fetus. In cases where severe asphyxia is diagnosed postnatally, available resuscitation methods can be employed in an attempt to protect the newborn’s life. However, it must be noted that the potential for such interventions remains limited in dairy and beef practices, underscoring the importance of preventing stillbirths. Consequently, this review focuses on the diagnostic possibilities and limitations related to the evaluation of fetal and neonatal vitality in dairy and beef practices.

**Abstract:**

Prior to initiating any obstetrical intervention for anterior or posterior presentation, it is imperative to emphasize the need for a precise and accurate diagnosis of fetal viability and to select the most appropriate approach for assistance. In uncertain cases, diagnostic tools such as ultrasonography, pulse oximeter, or measurement of acid–base balance or lactate concentration may be employed to confirm the diagnosis. In situations of severe asphyxia, a cesarean section is preferred over traction, even if the duration of asphyxia is less than 60 s, to maximize the likelihood of the survival of the fetus. Postcalving, several vitality scores have been proposed to evaluate the vigor of the newborn calf. Originally, four different clinical signs were recommended for assessing the well-being of newborn calves. Subsequently, five or more different clinical signs were recommended to evaluate vitality. However, despite the efforts for devising a practical tool to assess newborn calf vitality; a user-friendly and highly accurate instrument that can be used on farms remains elusive. Measuring the acid–base balance or lactate concentration may increase the diagnostic accuracy. It is critical to emphasize the importance of reducing the incidence of dystocia to mitigate the occurrence of severe asphyxia. In instances where asphyxia is unavoidable, adequate treatments should be administered to minimize losses.

## 1. Introduction

The profitability of a dairy farm depends greatly on the rate of calves being born alive and successfully reared to maturity [1,2]. Perinatal mortality, also referred to as stillbirth, pertains to the death of a mature fetal calf with a gestation period longer than 260 days that occurs during calving or within 24 to 48 h of postnatal life [3,4]. The incidence of perinatal mortality ranges from 3 to 10.3% [5], 2.4 to 9.7% [6], or 2 to 15% [7] across most countries. Despite considerable advancements in animal breeding, the prevalence of perinatal mortality remains high, particularly in Holstein-Friesian dairy farms [6,8,9,10]. In comparison, lower prevalence rates are mainly observed in relatively small farms with fewer than 65 dairy cows [5] or in dairy farms raising other breeds such as Norwegian Red or Swedish Red [6].

Over the past few decades, there has been an upward trend in stillbirth rates, especially in Holstein Friesian (HF) heifers. In Swedish, Dutch, and US HF heifer populations, the prevalence of stillbirth ranges between 10% and 13.2% [11,12,13,14,15,16], which has recently shown a static or declining trend [17]. However, there have been some encouraging outcomes in American [18] and Canadian dairy farms [19], with lower prevalence rates (<2%) of stillbirth. These findings highlight the importance of investigating the underlying factors contributing to perinatal mortality that have also been recently reviewed [20,21,22].

The etiology of stillbirth with a noninfectious origin is likely to be multifactorial, with direct and indirect asphyxia being the primary causes of death in most cases. Pathological changes were not detected in 58.3 to 75% of the calves that died during the perinatal period, as reported by several studies [23,24,25]. Furthermore, according to a recent review by Mee [26], the diagnostic rate of anoxia is highly variable between studies, ranging from approximately 10 to 80% of cases.

Fetal and maternal cannulation prior to calving allows for the monitoring of changes in the acid–base balance [27]. “These results indicate that the acid-base variables may start to move gradually in the direction of expressed respiratory and metabolic acidosis only after the amniotic sac and fetal feet appear in the vulva during the second stage of labor. However, in spontaneous calving, physiological respiratory or respiratory and metabolic acidosis is used to develop; therefore, it is essential to complete obstetrical assistance in time” [27]. The duration of survivable asphyxia depends on glycogen reserves in the heart [28]. In an experiment involving Hereford fetuses, the survival period was examined by the clamping of the umbilical cords. Four out of six fetuses survived after 4 min of anoxia, while none survived after 6 to 8 min of anoxia [29]. 

In dairy and beef practice, different methods are used to assess fetal and neonatal vitality as a measurement of the acid–base balance of fetuses or newborn calves is not feasible. Although portable acid–base analyzers are already available in the field, their present costs limit their daily use [30,31,32]. Concurrently, measuring lactate concentration using a handheld meter is a more cost-effective option [33,34]. 

The present review focuses on the determination of fetal and neonatal vitality of dairy calves to provide guidance for practitioners in choosing a suitable method to decrease fetal and/or neonatal mortality. Given the importance of reducing perinatal mortality rates in bovine cattle, recent scientific literature, including a published book [35] and several reviews [1,36,37,38], have emphasized the significance of this topic using different perspectives.

## 2. Determining Fetal Vitality during Stage Two of Calving

### 2.1. Determining the Clinical Signs of Vitality in Anterior and Posterior Presentation during Stage Two of Calving

The vitality of a bovine fetus during an obstetrical examination can be determined by initiating different reflexes in either the anterior (interdigital, bulbar, and swallowing reflexes) or posterior presentation (interdigital reflex, anal reflex, and pulse of the umbilical cord) [39]. 

As acidosis deepened (physiological, moderate, and severe acidosis), fetuses with anterior presentation (*n* = 180) failed to exhibit the interdigital reflex (88%, 65%, and 20%, respectively), bulbar reflex (95%, 91%, and 47%, respectively), and swallowing reflex (100%, 98%, and 80%, respectively). In the posterior presentation, a similar correlation between the palpable clinical signs of life (interdigital reflex: 52%, 25%, and 12%; anal reflex: 76%, 62%, and 50%; and the pulse of the umbilical cord: 95%, 87%, and 50%, respectively) and the acid–base balance could not be established due to the limited number of fetuses (*n* = 37). Nonetheless, there were 12 fetuses (5.5%) that were unable to initiate any reflexes but were still viable, as demonstrated by measuring the acid–base balance and providing obstetric assistance [39]. Therefore, before considering fetotomy, fetal heartbeat evaluation through B-mode/Doppler ultrasonography [40] and fetal pulse rate evaluation using a pulse oximeter [41] are crucial.

### 2.2. Measuring Fetal Acid–Base Values during Stage Two of Calving

In bovine practice, the measurement of the acid–base balance in fetal scalp blood is not as widely performed as in human practice, despite the commercial availability of portable acid–base analyzers. A study conducted by Bleul and Götz [31] compared 271 venous blood samples (obtained using the iSTAT analyzer) to reference methods and observed high correlations between acid–base parameters (r = 0.965–0.986), except for pO_2_ values (r = 0.817). Fetal blood samples for acid–base measurement can be collected by puncturing the *v. metacarpalis volaris superficialis* or *v. digitalis dorsalis communis III* before the onset of traction or the *a.* and *v. umbilicalis* before the onset of extraction during a cesarean (C-) section [1]. Capillary blood sampling is also a viable option for diagnosing fetal asphyxia [42].

Before and after birth, calves can be assigned to one of three groups according to their pH values, as suggested by Eigenmann et al. [43] and implemented by us [1,27]:

“Group 1: blood pH > 7.2–physiological acidosis = slight, combined respiratory and metabolic acidosis.

Group 2: blood pH 7.2–7.0–moderate acidosis = mild to expressed combined respiratory and metabolic acidosis.

Group 3: blood pH < 7.0, severe acidosis = severe, combined respiratory and metabolic acidosis” [43].

Immediately prior to parturition or obstetrical intervention, a substantial proportion of fetuses exhibited physiological metabolic acidosis, ranging from 57.6% to 80%. Moderate metabolic acidosis was observed in 20–24.9% of fetuses, while severe metabolic acidosis was present in 0–17.5% of cases (Table 1).

In contrast, Held et al. [47] evaluated the degree of prenatal acidosis by measuring the blood base excess (BE) value in 217 parturitions. The study revealed that 57.6% of the fetuses had physiologic (BE > −6.0 mmol/L), 24.9% had moderate (BE: −6.0 and −12.9 mmol/L), and 17.5% had severe acidosis (BE ≤ −13 mmol/L). Similar results were obtained using data collected from calves delivered by C-section, with 59.1% of the calves having physiological acidosis and 40.9% having moderate to severe acidosis [48], which indicated that both parameters (pH and/or BE) were suitable for classifying fetal acidosis prior to initiating calving assistance.

It is essential to mention that the detection of vigorous fetal movements during obstetric examinations or extraction of the fetus from the uterus during a C-section is indicative of the presence of severe metabolic acidosis. Therefore, assistance must be promptly completed. This is supported by our previous findings where four out of six calves showed fetal movements during C-section and were severely acidotic immediately after birth [48]. Additionally, the occurrence of fetal movements was confirmed in an experiment in which the umbilical cord was clamped [29]. The first movements occurred, on average, after 53 s postclamping of the umbilical cord (range, 10–105 s). Among the nine fetuses that died during the study, only one fetus did not show any movements after being subjected to ≥4 min of anoxia [29].

### 2.3. Measuring Fetal Oxygen Saturation during Stage Two of Calving

Electronic fetal heart rate (FHR) monitoring using cardiotocography is widely used to assess intrapartum hypoxia during labor in human practice [49]. In cases of nonreassuring cardiotocography, human fetal pulse oximetry accurately excludes moderate to advanced acidosis when the fetal pulse oximetry is <30% for at least 10 min. Additionally, it reduces the frequency of fetal blood analysis [50].

In bovine practice, continuous measurements of fetal oxygen saturation of arterial hemoglobin (FSpO_2_) via pulse oximetry (oxygen sensor) can also be utilized [1,51]. “An oxygen sensor designed for human babies was placed in the mouth against the mucosa of the hard palate. The accuracy of predicting asphyxia was the highest when the oxygen saturation was below for a period of at least 2 min. A cutoff value of <30% had the highest positive predictive value. Sporadic individual values <30% were not clinically significant as they might just represent physiological events at calving, e.g., the oxygen saturation can decrease temporarily during uterine contractions because of increased intrauterine pressure. At the same time, it was emphasized by Bleul and Kähn [51] that further studies are needed to determine whether an FSpO_2_ value of <30% over a minimum of 2 min is a valuable predictor of neonatal asphyxia in the cows”. Thus, rapid obstetrical assistance such as extraction or C-section would help prevent peripartal death of the calf by reducing the duration of asphyxia.

According to Kanz et al. [41], measuring pulse rate (>120 bpm for at least 2 min) by pulse oximetry during the last 25 min of calving could predict neonatal acidosis more accurately (area under the curve [AUC]: 0.764) than measuring fetal oxygen saturation (FSpO_2_ < 40% for at least 50% of the measurement: AUC = 0.613). The oximetric sensor was interdigitally fixed using a homemade latex cover. It was emphasized that improving the hardware of the device was necessary to obtain immediate results during the examination. Furthermore, improved fixation of the pulse oximeter to the fetus is essential for reducing the risk of intrapartal detachment.

### 2.4. Measuring Fetal Lactate Concentration during Stage Two of Calving

Recent advancements in dairy farming have led to the adoption of portable devices for measuring L-lactate concentration. Commercially available portable lactate meters, such as Lactate Scout and Biosen C line [33], as well as Accutrend Plus, i-STAT, Lactate Pro, and Lactate Scout [34], have been tested for their accuracy and found to be highly consistent with the reference methods (r = 0.98–0.99). A study by Karapinar et al. [34] reported that i-STAT had the highest sensitivity (100%) and specificity (98.6%) compared to the other three devices.

In human obstetrics, a fetal scalp blood lactate level of ≤5.4 mmol/L measured using the StatstripLactate^®^/StatstripXpress^®^ system is considered the cutoff for labor monitoring [52]. However, there is currently no established cutoff for bovine practice, and further research is needed in this area. 

### 2.5. Determining Fetal Heart Rate (FHR) during Stage Two of Calving

An FHR of 90–120 bpm prior to calving was established through transabdominal Doppler ultrasonographic examinations, which was calculated by visual inspection of paper recordings [53,54]. Subsequently, Breukelman et al. [40] used a computer-assisted analysis after an analog–digital conversion of the FHR, establishing the reference values (108.6 ± 1.9 bpm) for FHR two days before expected calving. Cardiotocographic recordings of 84 fetuses during the dilatation stage of parturition identified normocardia (heart rate: 80–155 bpm) and tachycardia (heart rate: >155 bpm), with a close correlation being found between heart rates and the degree of acidosis. Moderate or severe acidosis (BE < −8.9 mmol/L) was observed in all fetuses with tachycardia [55]. 

Calf births in poor condition were observed when decelerations (periodic decreases in FHR) occurred during the end of a contraction [56]. “In the acidotic group, the mean baseline FHR increased from 113.5 to 138.6 bpm during the last 55 min before birth, while in the normal group, it changed from 116.9 to 121.3 bpm. Decelerations occurred during uterine contractions in acidotic fetuses and in the majority of normal fetuses (12 of 16) during the expulsive stage of parturition; accelerations (periodic increase in FHR) were hardly recorded” [56]. The mean decrease in FHR during uterine contractions increased significantly toward birth, indicating that intrapartum continuous FHR measurements might provide additional information on the acidotic state of fetuses [57]. However, short-term measurements held no diagnostic value [56]. Therefore, at least 30 min of continuous recording is recommended for dairy cows [40]. 

Ultrasonic transit-time measurement of blood flow in the umbilical arteries and veins of the bovine fetus during stage II of calving facilitates direct and continuous measurement of umbilical blood flow volume per unit of time. It also allows for the investigation of the relationship between umbilical blood flow, uterine contractions, and fetal heart rate [58]. During uterine contractions, a decrease in blood flow is most likely caused by the compression of the umbilical vessels. Venous blood flow is more severely affected than arterial blood flow during muscular contractions due to the thinner walls and fewer muscle fibers of the umbilical veins than those of the umbilical arteries. In a study by Bleul et al. [58], lower umbilical arterial and venous blood flows in acidotic calves were found to be higher than those in nonacidotic calves during the last 30 min before birth. 

## 3. Diagnosis of Neonatal Vitality after Calving

### 3.1. Evaluating Clinical Signs of Vitality after Birth

Several parameters have been proposed as alternative vitality classification systems for field use after calving, including respiration and reflexes [59], time from birth until head-righting [29], until sternal recumbency [29,60,61], until the first apparent efforts to stand up [29,62,63], until the calf stands up [61,62,63,64]; until the first suckling [62,63,65,66,67], or a combination of attitude, vital signs, feeding behavior, and locomotion [68]. However, in most of these cases, the blood gases and acid–base parameters of newborn calves were not assessed, which can provide more accurate predictions of calf vitality [36]. 

In human obstetrical practice, a numerical scoring system has been suggested to assess a newborn baby’s clinical condition [69], which is based on heart rate, respiratory effort, muscle tone, reflex irritability, and skin color. Clinical signs are assessed for each variable, and a score of 0, 1, or 2 is assigned. The total score is used to calculate a final score (0–10), which is named Apgar points after the inventor [69].

In bovine practice, Mülling [70] was the first to adapt the human Apgar score (muscle tone and movement, reflex activity, respiration, and mucous membrane color) and proposed a neonatal status diagnosis in calves (Table 2), which was subsequently adopted by others [41,71,72,73,74,75]. Instead of the five clinical signs, four were assessed and scores were assigned ranging from zero to two. The higher the score, the more robust the calf: scores 7–8 indicate a healthy calf, scores 4–6 indicate a calf at risk, and scores ≤ 3 indicate a weak calf that is alive. In addition, healthy calves have better acid–base parameters than at-risk calves [41,71]. 

In contrast to previous recommendations, Mauer-Schweizer et al. [76] suggested the use of heart rate detection instead of assessing mucous membrane color for determining the health status of newborn calves (Table 3). Furthermore, they reported a definitive correlation between modified Apgar scores and acid–base parameters [77,78]. Palmer [79] adopted the same modified Apgar score system suggested by Mauer-Schweizer et al. [76]; however, it included the detection of reflex activity in response to nasal stimulation (no response, grimace, and sneeze/cough) and ear tickling (no response, weak ear flick, and ear flick/head shake).

Another modification of the Apgar score system developed by Mülling [70] was proposed by Born [80], who recommended evaluating the effect of splashing cold water on the calf’s head, eye and interdigital reflexes, respiration, and mucous membrane color after a C-section (Table 4). This system has been adopted by other researchers [39,81,82,83,84,85,86,87,88].

Essmeyer [89] suggested “evaluating the reaction to cold water on the head (no reaction; reduced, late reaction; lifting, shaking of the head), interdigital reflex (no reaction; pulling away slowly, weak; pulling back strongly, immediate reaction), mucosal membrane color (white; pale pink, cyanotic; pink) and respiration (absent; irregular frequency and intensity; regular frequency and intensity)” which was subsequently employed by Sorge et al. [90].

However, discrepancies between the measured pH values and modified Apgar scores were discovered by Born [80], who found agreement with Maurer-Schweizer and Walser [77]. Herfen and Bostedt [75] then demonstrated that the modified Mülling Apgar score [70] had only a marginal correlation with blood gas analysis results since 13 of 98 newborn calves had physiological pH values immediately after birth, but their modified Apgar scores were ≤3.

In contrast, Vollhardt [91] was the first to propose the use of five clinical parameters (respiration within 1 min, muscle tone and movement (head raising, extremities), reflex excitability (eyelid and claw reflex), conjunctival color, and suckling reflex) to assess the vitality of newborn calves (Table 5). Subsequently, Schulz and Vollhardt [92] found a strong correlation (r = 0.67) between vitality scores and venous blood pH values, while Gürtler et al. [93] found statistically significant relationships between vitality scores and venous blood pH (r = 0.69) and BE values (r = 0.61), as well as an indirect relationship with lactate levels (r = −0.66), following difficult births.

Later research conducted by Torres and Gonzales [94] introduced five distinct clinical criteria for evaluating neonatal vitality: responsiveness to exogenic stimuli, time to the raising of the head, sucking reflex, interest in the environment, and time needed to successfully stand up, and found a strong correlation with the acid-base status of newborn calves. A vitality score of 8–10 was considered indicative of a healthy calf, scores 6–7 suggested a calf at risk, and scores ≤ 5 indicated a weak calf that was still alive. Probo et al. [95] recommended the assessment of the following five variables: heart rate and rhythm (absent; irregular rhythm or <100 bpm; ≥100 bpm and regular rhythm), respiratory rate and rhythm (absent; irregular rhythm or <30 rpm; ≥30 rpm and regular rhythm), body tone (atonic, hypertonic, and sternal/active), the color of the mucous membranes of the eyes and mouth (hyperemic/cyanotic; pale; pink), and response to nasal and ear stimulation (absent response, grimace or weak response, and avoidance of stimulation). An index of ≥7 was considered normal. Subsequently, Vannucchi et al. [96] proposed the evaluation of five variables, such as “heart rate (absent; bradycardia/irregular: <120 bpm; normal/regular: 120–220 bpm), respiratory rate and effort (absent; irregular < 35 rpm; regular 35–90 rpm), muscle tone (flaccidity; some flexion; flexion), irritability reflex (absent; some movement; hyperactivity) and mucous color of eyes and gums (cyanotic; pale; normal)”. An index of ≥7 was considered normal. Concurrently, Kovács et al. [97] proposed the evaluation of newborn calf vitality through the following parameters: muscle tone (toneless, low, normal); erection of the head (head dropping; head requiring support; erected head); muscle reflexes (limbs extended; reduced number and intensity of reflectory movements; normal reflectory movements); heart rate (absent; bradycardia/irregular < 120 bpm; normal/regular 120–220 bpm), and sucking drive (absent; reduced; intensive). Higher scores indicate greater vigor.

Schuijt and Taverne [98] proposed evaluating newborn calf vitality based on the time from birth to the attainment of sternal recumbency (T-SR). “Calves were vital when they received routine care without medical treatment and survived seven days from birth without any symptoms of illness”. Nonvital calves were those who failed to meet the aforementioned criteria. “The mean ± SD T-SR values of the healthy calves were 4.0 ± 2.2 min (born spontaneously), 4.5 ± 3.1 min (C-section), 5.4 ± 3.3 min (usual traction), and 9.0 ± 3.3 min (forced traction), respectively. Calves delivered by forceful extraction had longer T-SR, more severe acidosis, recovered more slowly from acidosis, showed higher mortality, and exhibited trauma more frequently. A moderate correlation was reported between T-SR values and 10-min pH and BE values, while there was a weak correlation between T-SR and pCO_2_ values”.

It is worth noting that although the interval from birth to sternal recumbency is objective and usually short, it has less practicality during daily practice since, immediately after birth, an assisted calf has to be placed into sternal recumbency to help respiration [99]. Nevertheless, Uystepruyst et al. [99] have also measured the time between birth and sternal recumbency in newborn calves to evaluate the vitality delivered by elective C-section, as suggested by Schuijt and Taverne [98]. Similarly, Barrier et al. [62], Murray et al. [100], and Probo et al. [95] measured the time between birth and sternal recumbency in newborn calves to assess calf vigor.

Mee [101] suggested evaluating newborn calf vitality based on the assessment of the following variables: “the presence of meconium staining, peripheral edema, cyanosis of the mucous membranes, as well as heart and respiration rates, muscle tone, stimulation reflexes, rectal temperature, time to sternal recumbency and attempts to stand and suckle” (Table 6).

Murray [102] worked out a Dairy Calf VIGOR score based on ten physical exam parameters in five categories comprising the acronym VIGOR (Table 7):

“V” (visual appearance): evaluating the presence of meconium staining and the appearance of the tongue and head.

“I” (initiation of movement): detecting time to achieve certain postural behaviors. 

“G” (general responsiveness): pricking the nasal mucosa with a straw, pinching the tongue, and touching the eyeball. 

“O” (oxygenation): evaluating mucous membrane color and length of protruding tongue.

“R” (rates): measuring heart rate and respiratory rate. 

Villettaz Robichaud et al. [19] recently proposed modifications to the vigor score sheet (Table 7), which included the removal of the calf movement evaluation and the measurement of time to sternal recumbency and tongue length. Conversely, heart rate values were replaced with scores of one and two (Score 1: <80 bpm, Score 2: >100 bpm). “Time to sternal recumbency was treated as a continuous variable and analyzed separately from the vigor score, with shorter time indicating greater calf vigor”. However, as mentioned previously, an assisted calf must be placed into sternal recumbency immediately after birth if respiration initiation is delayed [99].

Homerosky et al. [103] examined additional parameters besides the traditional Apgar parameters (heart rate, respiration rate, and mucous membrane color) to identify newborn beef calves with acidosis, which included “meconium staining (visual assessment of hair coat and amniotic fluid: absent/present), tongue withdrawal (visual assessment of rostral aspect of the tongue: normal size within oral cavity/protruding or swollen) and suckle reflex (response to placing two fingers longitudinally in calf’s mouth: strong/weak). Nasal prick (reaction to pricking the nasal mucosa with straw: actively shakes head/minimal movement) and corneal reflex (response to touching conjunctiva with forefinger: complete blink/ incomplete blink)” were also evaluated and compared with the venous blood acid–base parameters obtained 10 min after birth. The study found that the traditional Apgar parameters, meconium staining, nasal prick test, and suckle and corneal reflex did not help identify newborn beef calves with acidosis. However, tongue withdrawal, calving ease, and parity may be useful in these assessments. 

On dairy farms, where obstetrician assistants are familiar with providing continuous surveillance, a simple vitality score system is necessary for immediate and accurate determination of neonatal vitality without requiring laboratory tests. Thus, immediate treatment can be provided whenever required [104]. The degree of neonatal vitality was evaluated based on muscle tone (active with head-righting within seconds), and in problematic cases, cardiac status was also considered. Newborn calves (*n* = 147) were examined immediately after birth and characterized as follows [104]:

“V-III.: Normal tonicity, head erect, and normal reflectoric movements

V-II.: Low tonicity, abdominal recumbency with head requiring support, and reduced number and intensity of reflectoric movements

V-I.: Toneless, head dropping, limbs extended, and cardiac activity present

V-0.: Toneless, head dropping, limbs extended, and cardiac activity absent [104]”

Significant differences were observed between the vitality scores of newborn calves assessed immediately after birth and their associated acid–base parameters, suggesting that this scoring system can provide valuable insights into the neonatal calf’s overall health status without the need for laboratory measurements. This approach allows for timely and appropriate administration of treatment, which is particularly important for Cesarean-derived calves, as these animals may take 2 to 3 min to lift their head and turn from lateral to sternal recumbency. In such cases, potential errors can be excluded when assessing the response of the calf to splashing cold water over its head [78,87] and/or turning them immediately into sternal recumbency [99]. In contrast, Schulz et al. [105] suggested evaluating suckling behavior as a criterion alone or as part of the modified Apgar score.

Despite several attempts to develop practical tools for assessing newborn calf vitality, a highly accurate and easy-to-use method suitable for on-farm use remains elusive [102].

### 3.2. Measuring Neonatal Acid–Base Values after Birth

Following spontaneous calving or obstetrical assistance with one assistant, a considerable proportion of newborn calves experience physiological metabolic acidosis, ranging from 39.7 to 80% [44,45,46]. Similarly, after C-section, acidosis is observed in 50% to 63.5% of cases [43,48,106]. The prevalence of moderate metabolic acidosis in newborn calves following spontaneous calving or obstetrical assistance ranges between 20% and 50%, with only a small proportion of cases (0–10.3%) being severe [44,45,46]. Additionally, after C-section, moderate metabolic acidosis prevalence rates are observed in 23.8% to 34.1% of cases, with severe metabolic acidosis ranging from 12.7% to 22.8% [43,48,106] (refer to Table 8). It is important to note that a C-section is a more cautious procedure than traction, particularly forced traction, which has been confirmed by other studies [43,48,93,107].

At birth, upon initiation of respiration, vasoconstriction ceases, and the accumulated acids enter circulation, resulting in a further decline in pH, and the acid–base variables are observed during the first 10 min postcalving. These metabolic reductions are more pronounced following a traction-assisted delivery [76,108,109,110] compared to a C-section [48,111]. Furthermore, posterior presentation deliveries exhibit more substantial decreases in metabolic values compared to anterior presentation deliveries [109,112].

In cases of severe respiratory metabolic acidosis, postnatal compensation for metabolic acidosis is relatively slower, taking between 1 and 6 h after birth, while respiratory acidosis may persist for up to 24 and 48 h postpartum [77,93,108,110,113]. Therefore, it is crucial to managing severe metabolic acidosis promptly by administering sodium bicarbonate or carbicarb infusion immediately after delivery [43,114].

### 3.3. Measuring Neonatal Oxygen Saturation after Birth

In human medicine [50], pulse oximetry is widely used to measure the percentage of oxygen saturation of arterial hemoglobin (SpO_2_) in a noninvasive and precise manner. Conversely, in bovine practice, the accuracy of a pulse oximeter in newborn calves was first examined by Uystepruyst et al. [115] by comparing SpO_2_ and arterial oxyhemoglobin saturation (SaO_2_), measured with a blood gas analyzer. Two transmission-type sensors of the pulse oximeter were placed on the proximal region of the tail (one on the dorsal surface and the second on the ventral surface), where the skin was nonpigmented and shaved. The study reported a highly significant correlation (r = 0.87) between mSpO_2_ (SpO_2_ values were recorded for 1 min and averaged) and SaO_2_ values [115]. 

Despite its accuracy and portability, as well as being a noninvasive, easy-to-use, and inexpensive technique suitable for dairy farm use, pulse oximetry cannot provide a precise absolute measurement of SaO_2_ in calves after birth due to the overestimation of SpO_2_ values when SaO_2_ values are lower. Additionally, its accuracy may be affected by several variables, including the pulse oximeter device, type of transducer, site of measurement, tissue perfusion, pigmentation of the site, ambient light (i.e., infrared heat lamps), or animal movement [115]. Despite these limitations, the use of pulse oximetry enables the objective evaluation of pulmonary function effectiveness in newborn calves during their transition to extrauterine life [115].

Kanz et al. [116] examined the accuracy of a pulse oximeter (Radius-7) after calving. The sensor was placed in the interdigital space of the front legs of the calves and fixed using a custom-made latex hoof cover. SpO_2_ values were compared with arterial SaO_2_ values measured using a portable blood gas analyzer as a reference, while pulse rates were measured and compared to the heart rate belt reference. The Spearman correlation coefficients for SpO_2_ and pulse rate were 93.8% and 97.7%, respectively. However, these results could be considered valid only for motionless calves with sternal recumbency. Therefore, purpose-built equipment is required for dairy practice.

### 3.4. Measuring Neonatal Lactate Concentration after Birth

Burfeind and Heuwieser [33] conducted a study to validate the accuracy of a hand-held meter (Lactate Scout) in measuring L-lactate concentration in newborn calves (age: 17 ± 12 days). They found a high correlation (r = 0.98) between the measurements obtained using the Lactate Scout and those obtained through reference laboratory methods. In another study, Homerosky et al. [103] found a strong negative correlation (r = −0.86) between L-lactate concentration and blood pH in neonatal calves 10 min after birth using a Lactate-Pro handheld meter. The authors concluded that lactate meters are practical and should be strongly considered for on-farm use, as they provide accurate estimates of blood pH. Similarly, Bleul and Götz [31] found a high correlation (r = 0.984) between blood L-lactate and pH (r > 0.95) measured by i-STAT and the reference laboratory methods in blood samples withdrawn immediately after birth. However, in 12 out of 46 newborn calves, the i-STAT lactate concentrations around the reference limit of 20 mmol/L could not be confirmed by measuring blood pH and BE values. Therefore, it was emphasized that there is a need for cautious interpretation of high L-lactate concentration due to suboptimal agreement between the higher L-lactate and reference laboratory levels. Simultaneously, if these calves were treated, they would not have had a detrimental effect since they also suffered from metabolic acidosis. Sorge et al. [88] reported that 8 out of 281 newborn calves had blood L-lactate concentrations of up to 19.8 mmol/L (Lactate-Pro) within 5 min after birth, despite a maximum Apgar score of eight. Therefore, in doubtful cases, it is suggested to confirm L-lactate concentration with the measurement of a new blood sample [103]. Although hand-held devices allow rapid, reliable, and accurate point-of-care blood analysis in dairy farms due to the high variability of L-lactate values, determination of the cut-off values is also warranted.

## 4. Future Perspectives

Currently, the main emphasis in obstetrical assistance must be placed on the prevention of asphyxia due to the lack of instruments that can reliably clear respiratory passages, maintain this state, and perform artificial respiration, as well as the insufficient competency-based skill of calving assistants to manage artificial respiration in the field. Although a calf aspirator/resuscitator for the suction of bronchial secretions is already available in bovine practice [1,117], intrauterine hypoxia during obstetrical assistance may develop depending on the degree of metabolic acidosis, organ injuries (hemorrhage), or meconium aspiration, which may increase the prevalence of neonatal mortality [118,119]. 

Reducing the need for calving assistance is the most critical breeding objective, particularly since calving assistance may shift the fetal acid–base balance toward acidosis [1]. However, in many cases, there are no visible clinical signs of calving onset, making it challenging to recognize. Using different sensors to predict the onset of calving may contribute to reducing stillbirth, delayed calving assistance, and its consequences in the field [2]. In cases of dystocia, the mode (traction or C-section) and the time of calving assistance should be chosen considering profitability factors and in a manner that minimizes the fetal acid–base balance towards acidosis [1,27,120]. Obstetric assistance should be initiated within 70 min after the appearance of the amniotic sac or 65 min after the appearance of hooves in the vulva [121]. Early intervention (within 15 min after the first sight of both front hooves in the vulva) does not necessarily increase the stillbirth rates, although the prevalence of injuries to the soft birth canal and retained fetal membranes can be increased [122]. Similarly, Villettaz Robichaud et al. [19] could not confirm the negative effect of early intervention on the stillbirth rate; nevertheless, sterile obstetrical lubricant was applied to the dam’s soft birth canal around the fetus before providing obstetrical assistance. Concurrently, the prevalence of injuries to the soft birth canal and retained fetal membranes has not been reported. Delayed obstetrical assistance can increase the rate of severely acidotic calves (pH < 7.0) (<2 h: 0%, 2–4 h: 19%, 4–7 h: 44%) [39]. Furthermore, while Vannucchi et al. [123] reported normal mean metabolic parameters even when the duration of calving was longer than 4 h, significant respiratory depression (altered lung gas exchange and delayed lung clearance) has also been reported [124].

Prior to conducting any obstetric traction, it is crucial to assess the degree of dilatation of the soft birth canal. If the dilatation is found to be insufficiently dilated, nonsurgical or surgical expansion techniques, such as episiotomy lateralis, must be performed [1]. Obstetric lubricants should also be utilized [18,19] to prevent tractions exceeding 2–3 min [46], as excessive traction can lead to rib and vertebral fractures [125]. 

A C-section should be performed if prolonged traction is expected to preserve the calf’s life and prevent maternal birth canal injuries. Recent research suggests that prior to deciding on the mode of calving assistance, the results of the acid–base balance or L-lactate measurements from blood samples should be considered [103,120]. The routinely applied complex treatment for asphyxiated newborn calves, which involves the initiation of respiration through physical stimulation or respiratory stimulants, providing oxygen/air supplementation, alleviation of pain and inflammation following dystocia, compensation of acidosis through buffer therapy, ensuring thermal support, and administering umbilical treatment, may reduce postnatal calf losses [36,101,117,126]. In addition to adequate treatment, special attention should be paid to the consumption and absorption of sufficient amounts of colostrum in asphyxiated newborn calves, as the lack of colostrum uptake is accompanied by an increased susceptibility to gastrointestinal disorders [19,37,68,100,127,128,129]. 

## 5. Conclusions

Several diagnostic methods are available to evaluate the clinical signs of fetal/neonatal vitality during and after calving, as well as to measure the acid–base balance or L-lactate concentrations. The use of ultrasonography to detect heart rate, or pulse oximetry to continuously measure fetal/neonatal oxygen saturation of arterial hemoglobin and heart rate on dairy or beef farms was observed. The implementation of these methods may contribute to recognizing and eliminating threats to the fetal/neonatal calf’s vitality in a timely manner. Hence, it is essential for farm management to select and apply these methods, considering current economic aspects, to prevent damage caused by dystocia, which frequently contributes to fetal/neonatal mortality. Nevertheless, even in today’s circumstances, primary emphasis must be placed on prevention, with medical treatments being a secondary consideration.

## Figures and Tables

**Table 1 animals-13-01081-t001:** Prevalence of physiological, moderately, and severely acidotic fetuses before starting obstetrical assistance.

Type of Obstetrical Assistance	Site of Blood Sampling before Calving	No. of Examined Fetuses (*n*)	pH > 7.2 *n* (%)	pH 7.2–7.0 *n* (%)	pH < 7.0 *n* (%)	References
Spontaneous or traction	*v. jugularis*	19	15 (78.9)	4 (21.1)	0	Eichler- Steinhauff [44]
Spontaneous or traction	*v. metacarpalis volaris superficial*	20	16 (80)	4 (20)	0	Mülling et al. [45]
Spontaneous or traction	*v. metacarpalis volaris superficial*	58	43 (74.1)	14 (24.1)	1 (1.7)	Szenci et al. [46]
Spontaneous, traction or C-section	*v. digitalis dorsalis communis III*	217 *	125 (57.6)	54 (24.9) ^a^	38 (17.5) ^b^	Held et al. [47]
Spontaneous, traction, or C-section	Fetal capillary blood	38	29 (76.3)	9 (23.7)		Bleul et al. [42]
C-section	*a.* or *v. umbilicalis*	44	28 (63.6)	10 (22.7)	6 (13.6)	Szenci and Taverne [48]

* Fetuses were grouped according to the BE values: physiological acidosis: BE > −6.0 mmol/L; moderate acidosis: BE: −6.0 and −12.9 mmol/L; severe acidosis: BE < −13 mmol/L [41]. ^a^ Four calves died within 24 h of birth (8%). ^b^ Fifteen calves died within 24 h of birth (51%).

**Table 2 animals-13-01081-t002:** Apgar score system modified for newborn calves by Mülling [70].

Parameters/Scores	0	1	2
Muscle tone and movement	Absence	Reduced	Spontaneous, active movement
Reflex activity	Being absent	Reduced	Fully available
Respiration	Being absent	Slow, irregular	Rhythmic, normal
Mucous membrane color	Bluish-white	Blue	Pink

Scores 7–8: normal, Scores 4–6: calf at risk, Scores 0–3: live weak.

**Table 3 animals-13-01081-t003:** Modification of Mülling’s Apgar score system by Maurer-Schweizer et al. [76].

Parameters/Scores	0	1	2
Respiration	Being absent	Slow, irregular,	Rhythmic, normal
Heart rate	Not measurable	<100, >150	100–150
Muscle tone and movement	Absence	Reduced	Active movement
Reflex activity	Being absent	Reduced	Fully available

Scores 7–8: normal, Scores 4–6: calf at risk, Scores 0–3: live weak.

**Table 4 animals-13-01081-t004:** Modification of Mülling’s Apgar score system by Born [80].

Parameters/Scores	0	1	2
Effect of splashing cold water on the head	Being absent	Reduced	Spontaneous, active movement
Eye and interdigital reflexes	Absence	One reflex is positive	Both reflexes are positive
Respiration	Being absent	Irregular	Rhythmic
Mucous membrane color	Bluish-white	Blue	Pink

Scores 7–8: normal, Scores 4–6: calf at risk, Scores 0–3: live weak.

**Table 5 animals-13-01081-t005:** Calf vitality score sheet recommended by Vollhardt [91].

Parameters/Scores	0	1	2
Respiration within 1 min	Absent	Irregular	Spontaneous
Muscle tone and movement (head raising, extremities)	Absent	Reduced	Vigorous
Reflex excitability (Eyelid and claw reflex)	Absent	Present	Very good
Conjunctival color	Bluish-white	White	Pink
Suckling reflex	Absent	Present	Vigorous

Scores 8–10: normal, Scores 5–7: calf at risk; scores < 5: live weak.

**Table 6 animals-13-01081-t006:** Calf vitality score sheet recommended by Mee [101].

Criterion	Good Vitality	Poor Vitality
Respiration	50–75 bpm and thoracic breathing	Gasping, primary apnea, irregular, abdominal breathing, bellowing, and secondary apnea
Hair coat appearance	Placental fluid-covered	Meconium-stained
Peripheral edema	None	Capital, lingual, or limb edema
Mucous membranes	Pink and normal capillary refill time	Cyanotic, pale, and slow capillary refill time
Response to reflex stimulation	Vigorous head shake, strong corneal suck, or pedal reflex	Weak or no response
Muscle tone	Active with head-righting within minutes	Inactivity and flaccid musculature
Heart rate	100–150 bpm and regular	>150 bpm followed by bradycardia (<80 bpm) and an irregular, decreasing rate
Rectal temperature	102–103 °F (39–39.5 °C) after calving declining to 101–102 °F (38.5–39 °C) by 1 h and stable	103–104 °F (39.5–40 °C) after calving declining to 101 °F (<38.5 °C) by 1 h and decreasing
Sternal recumbence	Achieved within 5 min	Prolonged lateral recumbence
Attempts to rise	Attempting to stand within 15 min Standing within 1 h	Delayed or no attempts to rise
Suckling	Commences within 2 h	Delayed or no attempts to suckle

**Table 7 animals-13-01081-t007:** Calf VIGOR score sheet recommended by Murray [102].

Visual Appearance				
Score	0	1	2	3
1. Meconium staining	Normal: no staining	Slight: around anal/tail head area	Moderate: extending over body	Severe: completely covered
2. Tongue/Head	Normal (no swelling, tongue not protruding)	Tongue protruding but not swollen	Tongue protruding and swollen	Head and tongue swollen, tongue protruding
**Initiation of Movement**				
3. Calf movement	Standing/walking	Attempts to stand	Sternal	On side, no efforts to rise
Taken within	0–30 min	30 min–1.5 h	1.5 h–3 h	>3 h
**General Responsiveness**				
4. Head shake in response to straw in nasal cavity	Shakes head vigorously	Moves head away	Twitches or flinches	Does not respond
5. Tongue pinch	Actively withdraws tongue	Attempts to withdraw	Twitches tongue	Does not respond
6. Eye reflex (in response to touching eyeball)	Actively blinks and closes eye	Slow to blink	Does not respond	-
**Oxygenation**				
7. Mucous membrane color	Bright Pink	Light Pink	Brick Red	White/blue
8. Length of tongue *	<50 mm	50–61 mm	>62 mm	-
**Rates**				
9. Heart rate **	80–100 bpm	>100 bpm	<80 bpm	-
10. Respiration rate ***	~24–36 rrpm	~24 rrpm	~>36 rrpm	-

* Measure from lips. This measurement was recorded only within 5 min of calving. ** Place hand on the calf’s chest. The pulse was recorded for 15 s and then multiplied by 4 to obtain beats per minute (bpm). *** View and/or place hand on the calf’s abdomen to count the approximate number of breaths for 15 s and multiply by 4 to get respiration rates per minute (rrpm). A lower score indicates greater vigor.

**Table 8 animals-13-01081-t008:** Prevalence of physiological, moderately, and severely acidotic newborn calves immediately after obstetrical assistance.

Type of Obstetrical Assistance	Site of Blood Sampling before Calving	No. of Examined Calves (*n*)	pH > 7.2 *n* (%)	pH 7.2–7.0 *n* (%)	pH < 7.0 *n* (%)	Reference
Spontaneous or traction	*v. jugularis*	15	12 (80)	3 (20)	0	Eichler-Steinhauff [44]
Spontaneous or traction	*v. metacarpalis volaris superficial*	20	15 (75)	5 (25)	0	Mülling et al. [45]
Spontaneous or traction	*v. jugularis*	58	23 (39.7)	29 (50) ^a^	6 (10.3) ^b^	Szenci et al. [46]
Spontaneous or C-section	*v. jugularis*	25	12 (48)	10 (40)	3 (12)	Herfen and Bostedt [74]
Spontaneous, traction, or C-section	*v. jugularis*	98	39 (39.8)	48 (49)	11 (11.2)	Herfen and Bostedt [75]
Spontaneous, traction, or C-section	*v. jugularis*	336	192 (57.1)	119 (35.4)	25 (7.4)	Leister [88]
Spontaneous, traction, or C-section	*v. jugularis*	38	26 (68.4)	12 (31.6)		Bleul et al. [42]
C-section	*v. jugularis*	57	30 (52.6)	14 (24.6)	13 (22.8) ^c^	Eigenmann et al. [43]
C-section	*v. jugularis*	44	22 (50)	15 (34.1)	7 (15.9) ^d^	Szenci and Taverne [48]
C-section	*v. jugularis*	126	80 (63.5)	30 (23.8)	16 (12.7) ^e^	Szenci et al. [106]

^a^ One calf died within 48 h of birth (3.4%). ^b^ Four calves died during calving, while two died within 48 h of birth (100%). ^c^ Nine calves died within 24 h of birth (69.2%). ^d^ Two calves died within 24 h of birth (28.6%). ^e^ Six calves died within 48 h of birth (37.5%).

## Data Availability

Data sharing not applicable.

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
