# Peer review of "Importance of Monitoring Fetal and Neonatal Vitality in Bovine Practices"

_animals, 2023, doi:10.3390/ani13061081_

Round 1
Reviewer 1 Report
This review highlights the importance to diagnose the fetal vitality accurately before starting obstetrical assistance as the evaluation of fetal and neonatal vitality plays an essential role in preventing the occurrence of stillbirth in dairy practices.
Introduction:
LINES 38-40 “Unfortunately, despite the speedy developments of animal 38 breeding, perinatal mortality is still very high (4 to 7%) and constitutes approximately 39 half of the total calf losses” [3-7]. I think that this sentence needs a more recent list of references, because the author refers to article published mainly fifty years ago: Anderson, D.C.; Bellows, R.A. Some causes of neonatal and postnatal calf losses. J. Anim. Sci. 1967, 26, 941 (Abstr.). 532 ;Voelker, H.H. Calf mortality by breeds, sexes, breeding and years. J. Dairy Sci. 1967, 50, 993 (Abstr.). 533 5. Szenci, O.; B. Kiss, M. Perinatal calf losses in large cattle production units. Acta Vet. Hung. 1982, 30, 85-95. 534;Mee, J.F. Perinatal calf mortality - recent findings. Irish Vet. J. 1991, 44, 80-85. 535 7. Vestweber, J.G. Respiratory problems of newborn calves. Vet. Clin. North Am. Food Anim. Pract. 1997, 13, 411-421. 536
I suggest more recent articles that focused upon the problem of still birth worldwide. In the following list you can find some more recent examples that could help to focus on the matter (but these are only some examples). The following rferences could be useful also for the sentence in the line 43-45.
Mee, J.F. Investigation of bovine abortion and stillbirth/perinatal mortality - similar diagnostic challenges, different approaches. Ir Vet J 73, 20 (2020). https://doi.org/10.1186/s13620-020-00172-0
Daniel Mota-Rojas et al., 2018. Is vitality assessment important in neonatal animals? 10.1079/PAVSNNR201813036
Bleul U. Risk factors and rates of perinatal and postnatal mortality in cattle in Switzerland. Livestock Sci. 2011;135:257–64.
Hoedemaker M, Ruddat I, Teltscher M, Essmeyer K, Kreienbrock L. Influence of animal, herd, and management factors on perinatal mortality in dairy cattle - a survey in Thuringia, Germany. Berl Münch Tierärztl Wochenschr. 2010;123:130–6.
Mee JF. Why do so many calves die on modern dairy farms and what can we do about calf welfare in the future? Animals. 2013;3:1036–57.
Wheelhouse N, Mearns R, Willoughby K, Wright E, Turnbull D, Longbottom D. Evidence of members of the Chlamydiales in bovine abortions in England and Wales. Vet Rec. 2015. https://doi.org/10.1136/vr.103075.
Murray CF, Leslie KE. Newborn calf vitality: risk factors, characteristics, assessment, resulting outcomes and strategies for improvement. The Veterinary Journal 2013;198:322–8. https://doi.org/10.1016/j.tvjl.2013.06.007.
Silva del Río N, Stewart S, Rapnicki P, Chang YM, Fricke PM. An observational analysis of twin births, calf sex ratio, and calf mortality in Holstein dairy cattle. Journal of Dairy Science 2007;90:1255–64. 40.
Bleul U. Respiratory distress syndrome in calves. Veterinary Clinics Food Animals 2009; 25:179–93.
Mee JF, Sánchez-Miguel C, Doherty M. Influence of modifiable risk factors on the incidence of stillbirth/perinatal mortality in dairy cattle. The Veterinary Journal 2014;199:19–23.
Mee JF, Berry DP, Cromie AR. Prevalence of, and risk factors associated with, perinatal calf mortality in pasture-based Holstein-Friesian cows. Animal 2008;2:613–29.
LINE 84 and 87: space prior and after the symbol =
LINE 356: space prior and after the symbol =
LINE 378 : delete the space among 65 and %
LINE 433: space after r and the symbol =
I suggest to update some of the references cited through the document as there are approximately 50 references published prior 1990 and many in the 60s.
Author Response
Responses to Reviewer Comments:
The author is grateful for the efforts of Reviewers in the evaluation of his manuscript. I appreciate your time spent with the review. Before answering the most important concerns, let me thank you for your valuable comments on the paper. I feel that Reviewers’ comments and recommendations were reasonable, and I tried to take them into account as far as possible while improving the manuscript. In my opinion, the activities of the reviewers have contributed significantly to the improvement of the quality of my paper. As you will see, I have made all the corrections required.
Reviewer I Comments
Rew#1: This review highlights the importance to diagnose the fetal vitality accurately before starting obstetrical assistance as the evaluation of fetal and neonatal vitality plays an essential role in preventing the occurrence of stillbirth in dairy practices.
AU: The author would like to thank Reviewer II for finding merit in the review manuscript.
Rew#1: LINES 38-40 “Unfortunately, despite the speedy developments of animal 38 breeding, perinatal mortality is still very high (4 to 7%) and constitutes approximately 39 half of the total calf losses” [3-7]. I think that this sentence needs a more recent list of references, because the author refers to article published mainly fifty years ago: Anderson, D.C.; Bellows, R.A. Some causes of neonatal and postnatal calf losses. J. Anim. Sci. 1967, 26, 941 (Abstr.). 532 ;Voelker, H.H. Calf mortality by breeds, sexes, breeding and years. J. Dairy Sci. 1967, 50, 993 (Abstr.). 533 5. Szenci, O.; B. Kiss, M. Perinatal calf losses in large cattle production units. Acta Vet. Hung. 1982, 30, 85-95. 534;Mee, J.F. Perinatal calf mortality - recent findings. Irish Vet. J. 1991, 44, 80-85. 535 7. Vestweber, J.G. Respiratory problems of newborn calves. Vet. Clin. North Am. Food Anim. Pract. 1997, 13, 411-421. 536
AU: It was changed according to the suggestion using new references:
Perinatal mortality (stillbirth) is the death of a mature fetal calf with longer than 260 days of gestation during calving or within 24 to 48 h of postnatal life [3,4]. It may range from 3 to 10.3% [5], from 2.4 to 9.7% [6], or from 2 to 15% [7] in most countries. Despite the speedy development in animal breeding, the higher prevalence rates are still very high, mainly occurring in Holstein-Friesian dairy farms [6,8-10]. In comparison, the lower prevalence rates mainly occur in relatively small farms with around >65 dairy cows or less [5] or in dairy farms raising other breeds like Norwegian Red or Swedish Red [6].
Rew#1: I suggest more recent articles that focused upon the problem of still birth worldwide. In the following list you can find some more recent examples that could help to focus on the matter (but these are only some examples). The following rferences could be useful also for the sentence in the line 43-45.
Mee, J.F. Investigation of bovine abortion and stillbirth/perinatal mortality - similar diagnostic challenges, different approaches. Ir Vet J 73, 20 (2020). https://doi.org/10.1186/s13620-020-00172-0
Daniel Mota-Rojas et al., 2018. Is vitality assessment important in neonatal animals? 10.1079/PAVSNNR201813036 of perinatal and
Bleul U. Risk factors and rates postnatal mortality in cattle in Switzerland. Livestock Sci. 2011;135:257–64.
Hoedemaker M, Ruddat I, Teltscher M, Essmeyer K, Kreienbrock L. Influence of animal, herd, and management factors on perinatal mortality in dairy cattle - a survey in Thuringia, Germany. Berl Münch Tierärztl Wochenschr. 2010;123:130–6.
Mee JF. Why do so many calves die on modern dairy farms and what can we do about calf welfare in the future? Animals. 2013;3:1036–57.
Wheelhouse N, Mearns R, Willoughby K, Wright E, Turnbull D, Longbottom D. Evidence of members of the Chlamydiales in bovine abortions in England and Wales. Vet Rec. 2015. https://doi.org/10.1136/vr.103075.
Murray CF, Leslie KE. Newborn calf vitality: risk factors, characteristics, assessment, resulting outcomes and strategies for improvement. The Veterinary Journal 2013;198:322–8. https://doi.org/10.1016/j.tvjl.2013.06.007.
Silva del Río N, Stewart S, Rapnicki P, Chang YM, Fricke PM. An observational analysis of twin births, calf sex ratio, and calf mortality in Holstein dairy cattle. Journal of Dairy Science 2007;90:1255–64. 40.
Bleul U. Respiratory distress syndrome in calves. Veterinary Clinics Food Animals 2009; 25:179–93.
Mee JF, Sánchez-Miguel C, Doherty M. Influence of modifiable risk factors on the incidence of stillbirth/perinatal mortality in dairy cattle. The Veterinary Journal 2014;199:19–23.
Mee JF, Berry DP, Cromie AR. Prevalence of, and risk factors associated with, perinatal calf mortality in pasture-based Holstein-Friesian cows. Animal 2008;2:613–29.
AU: This suggestion was accepted and used the relevant references from the list and new ones were also added.
Rew#1: LINE 84 and 87: space prior and after the symbol =
LINE 356: space prior and after the symbol =
LINE 378 : delete the space among 65 and %
LINE 433: space after r and the symbol =
AU: They were changed as requested.

Reviewer 2 Report
General comments
This is a thorough review of the monitoring practices of bovine fetal and neonatal vitality. The paper is very interesting and lists in detail all the literature on the subject. However, there is ambiguity in the wording of the information—grammatical errors that render the text unreadable, mainly from lines 37 to 209 and it is needed to be rephrased.
detailed notes:
lines 11-13, 19, 20, 29, 31, 94, 129, 366, 469, 496: «we diagnose….. we must be….» please rephrased
line 113-116 : please add reference
line 135-138: it is not clearly understandable. which 7 fetuses? above mentioned 6. please clarify and add reference.
Line 183-187: it is not clearly understandable. Please clarify and add reference.
Line 187-199: Which study, [52] or [53]? Please clarify and add reference.
Line 200: «Transit-time ultrasonography» the terminology is not correct « stage 2 of labor»? please rephrased.
Line 207-208: «Umbilical arterial and venous blood flow in acidotic calves was lower
than in non-acidotic calves…» I suggest ‘ In a study of….has been found lower than….’
Line 351 and 462: «large dairy farms» please rephrased , I suggest ‘ farm animals with capacity of x animals’
Line 354-356: it is not clearly understandable.
Line 363-366: please add reference
Line 377-379, 380-381, 382-383: please add reference.
Line 403: please add reference for human medicine
Line 405: «were the first to evaluate the accuracy….» please rephrased.
Author Response
Responses to Reviewer Comments:
The author is grateful for the efforts of Reviewers in the evaluation of his manuscript. I appreciate your time spent with the review. Before answering the most important concerns, let me thank you for your valuable comments on the paper. I feel that Reviewers’ comments and recommendations were reasonable, and I tried to take them into account as far as possible while improving the manuscript. In my opinion, the activities of the reviewers have contributed significantly to the improvement of the quality of my paper. As you will see, I have made all the corrections required.
Reviewer II Comments
This is a thorough review of the monitoring practices of bovine fetal and neonatal vitality. The paper is very interesting and lists in detail all the literature on the subject. However, there is ambiguity in the wording of the information—grammatical errors that render the text unreadable, mainly from lines 37 to 209 and it is needed to be rephrased.
AU: The author would like to thank Reviewer II for finding merit in the manuscript.
Rew#2: detailed notes:
lines 11-13, 19, 20, 29, 31, 94, 129, 366, 469, 496: «we diagnose….. we must be….» please rephrased
line 113-116 : please add reference
line 135-138: it is not clearly understandable. which 7 fetuses? above mentioned 6. please clarify and add reference.
Line 183-187: it is not clearly understandable. Please clarify and add reference.
Line 187-199: Which study, [52] or [53]? Please clarify and add reference.
Line 200: «Transit-time ultrasonography» the terminology is not correct « stage 2 of labor»? please rephrased.
Line 207-208: «Umbilical arterial and venous blood flow in acidotic calves was lower
than in non-acidotic calves…» I suggest ‘ In a study of….has been found lower than….’
Line 351 and 462: «large dairy farms» please rephrased , I suggest ‘ farm animals with capacity of x animals’
Line 354-356: it is not clearly understandable.
Line 363-366: please add reference
Line 377-379, 380-381, 382-383: please add reference.
Line 403: please add reference for human medicine
Line 405: «were the first to evaluate the accuracy….» please rephrased.
AU: All of these requests were accepted and changed accordingly.

Round 2
Reviewer 1 Report
Dear Author,
thank you for the improvement of your manuscript with new updated references and for editing the paper where required.
I think that in some passages, such as for example in the Simple Summary, the English is hard to follow and probably an English editing of the manuscript could be useful to check the grammar and phrasing thus improving its readability and quality. In any case, this is only a minor suggestion but I think that it could really improve your manuscript and enhance its content.
Author Response
Dear Reviewer,
please note that the whole manuscript was edited by EDITAGE. Hopefully, it will be OK.
Yours sincerely,
Otto Szenci
